# Bioinformatic Analysis Predicts a Novel Genetic Module Related to Triple Gene and Binary Movement Blocks of Plant Viruses: Tetra-Cistron Movement Block

**DOI:** 10.3390/biom12070861

**Published:** 2022-06-21

**Authors:** Sergey Y. Morozov, Andrey G. Solovyev

**Affiliations:** 1A. N. Belozersky Institute of Physico-Chemical Biology, Moscow State University, 119992 Moscow, Russia; solovyev@belozersky.msu.ru; 2Department of Virology, Biological Faculty, Moscow State University, 119234 Moscow, Russia; 3Institute of Agricultural Biotechnology, Russian Academy of Agricultural Sciences, 127550 Moscow, Russia

**Keywords:** RNA genome, plant virus, movement gene module, benyviruses, evolution, RNA helicase, membrane-embedded protein, double-stranded RNA binding protein

## Abstract

Previous studies have shown that the RNA genomes of some plant viruses encode two related genetic modules required for virus movement over the host body, containing two or three genes and named the binary movement block (BMB) and triple gene block (TGB), respectively. In this paper, we predict a novel putative-related movement gene module, called the tetra-cistron movement block (TCMB), in the virus-like transcriptome assemblies of the moss *Dicranum scoparium* and the Antarctic flowering plant *Colobanthus quitensis*. These TCMBs are encoded by smaller RNA components of putative two-component viruses related to plant benyviruses. Similar to the RNA2 of benyviruses, TCMB-containing RNAs have the 5′-terminal coat protein gene and include the RNA helicase gene which is followed by two small overlapping cistrons encoding hydrophobic proteins with a distant sequence similarity to the TGB2 and TGB3 proteins. Unlike TGB, TCMB also includes a fourth 5′-terminal gene preceding the helicase gene and coding for a protein showing a similarity to the double-stranded RNA-binding proteins of the DSRM AtDRB-like superfamily. Additionally, based on phylogenetic analysis, we suggest the involvement of replicative beny-like helicases in the evolution of the BMB and TCMB movement genetic modules.

## 1. Introduction

All triple gene block (TGB)-containing viruses are represented by a large variety of plant RNA viruses within the orders *Martellivirales* (family *Virgaviridae*), *Tymovirales* (families *Alphaflexiviridae* and *Betaflexiviridae*) and *Hepelivirales* (family *Benyviridae*). They have a positive-sense, single-stranded genome consisting of one to four RNA segments [1,2,3]. The TGB-encoded movement proteins, referred to as TGB1, TGB2 and TGB3, perform the directed transport of viral genomes to and through plasmodesmata (PD) into adjacent non-infected cells, the function unique for viruses infecting plants having tissues built up of simplistically connected cells. TGB1 contains the protein domain of RNA helicase of superfamily 1 (SF1), whereas TGB2 and TGB3 are the small membrane-associated proteins and contain highly hydrophobic amino acid segments [2]. Recently, the binary movement block (BMB), which is related to TGB, was found in plant viruses of the family *Kitaviridae* (order *Martellivirales*). This gene module includes only two genes termed BMB1 and BMB2. Although the pairs of BMB1/TGB1 and BMB2/TGB2 proteins are quite similar in structural, functional and phylogenetic aspects [4,5,6,7,8], the BMB gene module lacks a counterpart of TGB3. Importantly, recent data showed that some viruses from genera *Potexvirus* and *Carlavirus* encode no TGB3 proteins [4]. These observations support the hypothesis that the TGB3 gene could be a less essential component of TGB, or an accessory TGB cistron. Conceivably, TGB3 could be evolved as an additional cistron if the first transport module of this type that appeared in virus genomes had two genes like BMB or, alternatively, could be eliminated during the evolution of the earliest movement modules of the TGB type [4,5,6]. Taking into account these considerations, the identification of new TGB/BMB-like gene modules in lower land plants could shed new light on the early steps of TGB/BMB evolution. So far, only a single TGB-containing virus-like transcriptome contig has been revealed in non-seed plants, namely, bird’s-nest fern *Asplenium nidus*. The RNAseq-derived transcriptomic data obtained for plants often contain contigs corresponding to RNA viruses infecting these particular hosts, therefore, as suggested earlier, the TGB-containing the transcriptomic contig found for *A. nidus* corresponds to a genomic RNA of a previously unknown plant virus [4]. This fern virus shows gene arrangements and sequence similarities indicating its relatedness to benyviruses and *Nicotiana velutina mosaic virus* [4].

In our recent paper, we drew attention to partial transcriptome assemblies of the Antarctic flowering plant *Colobanthus quitensis*, where a probable evolutionary early variant of TGB was found (Cq-TGB) [6]. Importantly, the Cq-TGB1 protein sequence is more similar to BMB1 in comparison with TGB1 proteins (Figure 1). Moreover, the central hydrophilic region of the Cq-TGB3 protein located between two transmembrane sequence segments shows clear amino acid sequence similarity to the Cq-TGB2 protein and exhibits conservation of most amino acid residues invariant in other TGB2 proteins [6]. These data suggest that Cq-TGB could arise from a BMB-like module by duplication of the hydrophobic protein gene. As the Antarctica coast flora has been isolated from the rest of the world for approximately 20 million years [9], Cq-TGB may be considered as one of the ancient movement gene modules [6]. These observations have prompted us to carry out further searches of available sequence databases in an attempt to find more viral gene blocks related to TGB/BMB modules in transcriptomes of non-seed plants and extant land plants.

In this paper, we predicted new virus-like RNA assemblies (VLRAs) in the NCBI TSA and SRA databases and identified a novel plant virus movement gene module in the transcriptome data of two plant species. This gene module could be classified as structurally and evolutionary related to BMB and TGB. Additionally, we presented new data supporting the hypothesis that the movement gene modules related to TGB could initially originate in benyvirus-like replicons.

## 2. Materials and Methods

Virus nucleotide and protein sequences were collected from the NCBI database. Assembled viral genomes were mainly extracted from NCBI database. The sequence comparisons were carried out using the BLAST algorithm (BLASTn and BLASTp) at the National Center for Biotechnology Information (NCBI). Open reading frames (ORFs) were identified using the ORF Finder program (http://www.bioinformatics.org/sms2/orf_find.html and https://www.ncbi.nlm.nih.gov/orffinder/, accessed on 26 May 2022). Gene translation and prediction of deduced proteins were also performed using ExPASy (http://web.expasy.org/translate/ accessed on 26 May 2022). Conserved motif searches were conducted in CDD (http://www.ncbi.nlm.nih.gov/Structure/cdd/wrpsb.cgi, accessed on 26 May 2022) databases. Coiled-coil protein regions were predicted using Waggawagga software (https://waggawagga.motorprotein.de/ accessed on 26 May 2022) [10]. To assemble the full-length plant VLRAs, transcriptome sequencing data for *D.*
*scoparium* and C. *quitensis* SRA experiments linked to the TSA projects were downloaded using fastq-dump tool of NCBI SRA Toolkit (http://ncbi.github.io/sra-tools/ accessed on 26 May 2022). Reads quality was checked with FastQC (https://www.bioinformatics.babraham.ac.uk/projects/fastqc/ accessed on 26 May 2022). De novo assembly of VLRAs coding for TCMB modules was carried out using SPAdes (http://cab.spbu.ru/software/spades/ accessed on 26 May 2022) [11] in “RNA mode”.

Phylogenetic analysis was performed with “Phylogeny.fr” (a free, simple-to-use web service dedicated to reconstruction and analysis of phylogenetic relationships between molecular sequences) by constructing maximum likelihood phylogenetic trees (http://www.phylogeny.fr/simple_phylogeny.cgi accessed on 26 May 2022). Bootstrap percentages received from 1000 replications were used. Genome sequences of representative viruses in different beny-like viruses were downloaded from the GenBank database.

All web services used for sequence analysis in this paper are accessible on 19 June 2022.

## 3. Results

### 3.1. Novel Movement Gene Module in the Virus-Related Plant Transcriptoms: Tetra-Cistron Movement Block

We undertook a systematic analysis of RNA-seq datasets from *Viridiplantae,* available in the NCBI open-access TSA and SRA, at the end of March 2022. Our TBLASTn search of NCBI TSA data collection, using Cq-TGB encoded helicase (accession GCIB01126289) as a query, resulted in the identification of a new partial VLRA (accession HANF01089872) in moss *Dicranum scoparium* (family *Dicranaceae*, order *Dicranales*) encoding a protein showing similarity to Cq-TGB1 (Figure 1). Using transcriptome sequencing data for the *D.*
*scoparium* (SRA accession ERX3824048), we further assembled a nearly full-length sequence of the contig (Ds-VLRA2) comprising 3996 nt excluding the poly(A) tail (Appendix A). The open reading frame (ORF) prediction at ExPASy (ESTscan) showed that the contig contains six ORFs flanked by a 5′ untranslated region (5′ UTR, at least 183 nt) and a 3′ UTR (165 nt) (Figure 2). Ds-VLRA2 exhibits a gene arrangement quite similar to that in RNA 2 of benyviruses [12]. Indeed, this RNA encodes the 5′-terminal coat protein (CP) gene (ORF1) and a TGB-like module (Figure 2). Although the CP of Ds-VLRA2 (220 aa in length) has only marginal amino acid similarity to the CPs of genus *Benyvirus*, it shows significant similarity to other tobamovirus-like CPs, namely, those of the tobacco rattle virus CP (genus *Tobravirus*; family *Virgaviridae*) (ABE27877, 31% identity, E-value 1 × 10^−11^), and the pea early-browning virus CP (genus *Tobravirus*; family *Virgaviridae*) (CAA07067, 29% identity, E-value 2 × 10^−11^). 

The predicted ORF2 of Ds-VLRA2 encodes a protein of 301 aa residues in length (Figure 2). BLASTn and BLASTx analyses showed that the ORF2 protein has no evident sequence similarity to Cq-encoded proteins or other known virus polypeptides (data not shown). The NCBI Conserved Domain Database (CDD) analysis identified that the ORF2 protein could contain a possible domain related to the SMC superfamily (Accession No. cl34174, E-value 6.56 × 10^−3^). The SMC (structural maintenance of chromosomes) domain is found in the proteins that bind DNA and act in organizing and segregating chromosomes [13]. Additional protein domain analyses using ExPASy ProtScale software (https://web.expasy.org/protscale/ accessed on 26 May 2022, SIB Swiss Institute of Bioinformatics, Lausanne, Switzerland) predicted a coiled-coil motif located at amino acid positions 136–185 (Figure 3) and two highly hydrophobic sequences positioned at residues 45–63 and 279–299 in the ORF2 protein.

The NCBI CDD analysis of the predicted ORF3 protein (120 aa in length) revealed a conserved DSRM_AtDRB-like domain (cd198178) assigned to the superfamily cl00054 in the NCBI conserved domain database. Therefore, the Ds-VLRA2 ORF3 protein was named viral DRB (vDRB). The DSRM protein domain superfamily is a well-known protein structural motif of 65–70 aa in length that adopts an alpha-beta-beta-beta-alpha fold and binds double-stranded RNAs (dsRNAs) of various origins. The DSRM_AtDRB-like domain is found in a group of *Arabidopsis thaliana* dsRNA-binding proteins termed AtDRB1-AtDRB5. Members of this group, which contain two DSRM domains, bind dsRNA and are involved in RNA-mediated silencing [14,15] and/or dsRNA-triggered immunity against viruses [16]. The vDRB protein encoded by Ds-VLRA2 contains a single DSRM domain showing conservation of key residues specific for DSRM_AtDRB-like proteins (Figure 4). Interestingly, AtDRB-like proteins are encoded not only by flowering plants but also by representatives of lower vascular plants (*Lycopodiopsida*), mosses (*Bryophyta*) and liverworts (*Marchantiophyta*). Moreover, these proteins can be revealed in present-day Charophyte algae (members of *Zygnemophyceae* and *Charophyceae* families), which are assumed to be the closest relatives of land plants descendent of the organisms that took part in the initial colonization of terrestrial habitats (Figure 4). Importantly, as revealed by hydrophobicity plot analysis (https://web.expasy.org/protscale/ accessed on 26 May 2022), the Ds-VLRA2 vDRB protein, unlike AtDRB1-AtDRB5, contains a highly hydrophobic segment at the N-terminus (data not shown).

The predicted ORF4 in Ds-VLRA2 overlaps the ORF3 by 60 nucleotides (Figure 2) and encodes a protein of 361 residues containing motifs characteristic of helicases of the SF1 superfamily and showing obvious similarity to BMB1 helicases (Figure 1). ORF4 is followed by overlapping ORF5 and ORF6 (Figure 2), which code for small hydrophobic proteins having putative transmembrane domains and a central hydrophilic region and related to TGB2/BMB2 proteins [5,6]. The protein sequence similarity of the encoded proteins and general organization of the gene block represented by ORFs 4-6 of Ds-VLRA2 resemble those of TGB and the related Cq-TGB-like module (accession GCIB01126289). Moreover, Ds-ORF5 and Ds-ORF6 proteins show some similarity in the central hydrophilic regions to their counterparts in Cq-TGB [6] (Figure 2 and Figure 5).

Taking into account the similarity of Ds-VLRA2 TGB and Cq-TGB, we further analyzed whether the previously assembled *C. quitensis* TGB-containing VLRA [6] was incomplete and could be extended into the 5′-terminus direction. Our de novo assembly of transcriptome sequencing data for the *C.*
*quitensis* SRA data set (SRX814890) resulted in probable near-full-length contig (Cq-VLRA2) of 4718 nt in length excluding the poly(A) tail (Appendix A). The resulting sequence was confirmed using additional incomplete contigs from the TSA database (GCIB01147888, GCIB01142942, GCIB01133924 and GCIB01126289). The ORF prediction at ExPASy showed that the contig contains seven ORFs flanked by a 5′-untranslated region (5′-UTR, at least 259 nt) and a 3′-UTR (213 nt). The 5′-terminal ORF1 encodes CP (164 aa in length) (Figure 2) showing a similarity to the wheat stripe mosaic virus CP (genus *Benyvirus*) (YP009553316, 27% identity, E-value 1 × 10^−4^), and the sorghum chlorotic spot virus CP (genus *Furovirus*; family *Virgaviridae*) (NP659022, 29% identity, E-value 3 × 10^−4^). The next ORF2 represents a read-through domain of CP as it was reported for benyviruses (Figure 2) [12]. Among the three described types of nucleotide contexts that determine the read-through of the leaky translation terminators, ORFs1/2 contain the type I motif containing a UAG codon, which is followed by the consensus motif CARYYA (where R is a purine and Y is a pyrimidine). This mechanism of translation is also used in tobamovirus replicase genes [17,18].

ORF2 is followed by an intergenic region of 73 nucleotides in length and an ORF3, which codes for a small protein with a charged N-terminal half and a cysteine-rich C-terminal region (Figure 2 and Figure 6). This cysteine-rich protein (CRP) shows no sequence similarity to the benyvirus RNA2-encoded CRP [12], and its cysteine-rich region is marginally similar to double zinc finger motif-containing module of FYVE domain involved in mRNA transport to endosomes [19].

The NCBI BLAST analysis showed that the predicted Cq-VLRA2 ORF4 protein is quite similar to Ds-ORF3 (vDRB) protein and contains a single DSRM with signatures specific for DSRM_AtDRB-like proteins (Figure 4) and the N-terminal hydrophobic segment (data not shown). Cq-ORF4 is followed by an overlapping ORF5 encoding protein of 355 amino acids in length (Figure 2). The Cq-ORF5 protein (earlier named Cq-TGB1, see above) possesses motifs characteristic of helicases of the SF1 superfamily and shows significant similarity to Ds-ORF4 protein (Ds-TGB1) and BMB1 helicases (Figure 1). Cq-ORF4 is followed by overlapping ORFs 5 and 6 (Figure 2) encoding TGB2 and TGB3 proteins with two putative transmembrane domains [6] and a central hydrophilic region related to Ds-ORF5/6 proteins (Figure 5).

A general view on the predicted organization of Ds-VLRA 2 and Cq-VLRA 2 strongly suggests that: (i) both virus-like RNAs have considerable similarity to the benyvirus TGB-containing RNA 2. Indeed, RNA 2 of the type benyvirus *Beet necrotic yellow vein virus* (BNYVV) has six ORFs, namely, the CP gene terminated by a leaky stop codon, the CP read-through protein gene, the TGB and the cistron coding for a cysteine-rich protein having a silencing suppressor activity (Figure 2) [12,20]; (ii) these RNAs include a conserved module of four overlapping genes, which is proposed to be named the “Tetra-Cistron Movement Block” (TCMB). In comparison with the TGB and BMB modules, TCMB includes an additional 5′-terminal ORF, which overlaps the downstream gene and codes for the vDRB protein. It should be noted that the cellular DRBs were shown to be incorporated into virus-specific replication membrane compartments which are often located at the PD orifice and can be involved in virus cell-to-cell movement [21,22,23,24].

### 3.2. Proposed General Organization of TCMB-Containing Plant Virus Genomes

Assuming the similarity of Ds-VLRA 2 and Cq-VLRA 2 to benyvirus RNA 2 (Figure 2) in gene organization, we performed a search of the NCBI *Dicranum scoparium* TSA database in an attempt to find Ds-RNA1 expected to code for virus replicase as in the case of BNYVV. As an initial query, we used a 150 amino acid-long segment of the BNYVV replicase (GDD domain). A BLAST search revealed a single TSA contig (HANF01090670) of 305 nucleotides in length which codes for a protein segment containing a GDD motif typical for the RNA-dependent RNA polymerase (RdRp) domain and having more than 60% protein identity to BNYVV replicase protein (data not shown). To assemble the expected Ds-RNA 1, the transcriptome sequencing data for *D.*
*scoparium* SRA experiment ERX3824048 linked to the TSA project were used. The assembled full-length sequence of the contig comprised 6624 nt, excluding the poly(A) tail. ORF prediction showed that the contig contains a single cistron encoding viral replicase (Appendix A) flanked by a 5′-UTR (at least 78 nt) and a 3′ UTR (237 nt). Importantly, pairwise BLASTN analysis of the 3′-untranslated regions from Ds-VLRA 1 and Ds-VLRA 2 indicated a significant degree of sequence conservation among them and strongly suggested that both moss VLRAs are indeed the two components of a single virus genome (Figure 7).

ORF1 protein of Ds-VLRA 1 (Appendix A) contains four conserved domains: a viral methyltransferase domain (MTR, pfam01660, amino acid positions 440–625, E-value 2.58 × 10^−^^6^); a viral helicase 1 domain (HEL, pfam01443, amino acid positions 939–1179, E-value 4.70 × 10^−22^); papain-like proteinase domain (PROT, pfam05415, positions 1333–1408, E-value 6.97 × 10^−6^), and RdRp core motif with GDD signature (pfam00978, amino acid positions 1698–2045, E-value 2.42 × 10^−14^). The MTR domain is known to be conserved in a wide range of single-stranded RNA viruses, including the orders *Martellivirales*, *Tymovirales* and *Hepelivirales* [25]. All replicases in the members of these orders also encode HEL and RdRp domains containing typical motifs [26] conserved also in the ORF1 protein of Ds-VLRA 1. Although the protease domain is not common for the above-mentioned replicases, Ds-VLRA 1 encodes a protein domain PROT with a similarity to benyvirus protease (data not shown), which is conserved in most benyviruses and required to produce mature replicase proteins by proteolytic self-cleavage [27].

We further used encoded amino acid and nucleotide sequences of Ds-VLRA1 as queries for searches of *C.*
*quitensis* SRA data (SRX814890) in order to assemble a Cq-RNA1 complete nucleotide sequence. However, only a rather short nucleotide sequence encoding a part of RdRp domain (including the GDD signature), which showed significant similarity to the Ds-VLRA1 protein and moderate similarity to benyvirus replicases, has been assembled (data not shown). This failure to identify the full-length Cq-RNA1 could likely result from the insufficient coverage of the used sequencing data.

## 4. Discussion

The transport gene module found in RNAs of viral origin identified in the transcriptomes of moss *D. scoparium* and Antarctic flowering plant *C. quitensis* represent a conserved array of four overlapping genes named the “Tetra-Cistron Movement Block” (TCMB), which is related to the well-characterized transport modules TGB and BMB, consisting of three and two genes, respectively [1,2,5,7]. The characteristic feature of TCMB is the presence of an additional 5′-terminal ORF, which overlaps the downstream gene for a TGB1/BMB1-related protein and codes for a protein containing a dsRNA-binding domain of the DSRM_AtDRB family (Figure 4). This protein was termed vDRB (viral dsRNA-binding protein). Previously, domains of the DSRM_AtDRB family have never been found in virus-encoded proteins; however, plant viruses have been reported to encode proteins with the dsRNA bonding domains of other families. For example, *Sweet potato chlorotic stunt virus* (SPCSV, a crinivirus) encodes an RNAse III with conserved dsRNA-binding domain (cd10845) [28,29], whereas the B2 protein of *Flockhouse virus* (FHV), which can infect both insect and plant cells, contains a dsRNA-binding domain of another type (superfamily cl12995) [30]. Our searches of the most recent plant virus sequence database (https://riboviria.org/ accessed on 26 May 2022) revealed additional dsRNA-binding domains of superfamily cl00054 in ORF2 of beny-like RNAs encoding viral replicase as an ORF1. These include a motif of the DSRM_RNAse_III_family (cd10845; contigs ND_248078 and ND_226011), as well as some other motifs (contigs ND_098871, ND_246367, ND_143889 and ND_142202). These observations demonstrate that dsRNA-binding proteins are encoded by a variety of plant viruses, including those with beny-like replicons.

In many viruses infecting plant and non-plant hosts, dsRNA-binding domains are found in proteins that play a role in subverting host antiviral responses [31,32]. Particularly, the dsRNA-binding proteins of FHV and SPCSV are shown to be suppressors of RNA silencing [28,30]. Therefore, the vDRB proteins encoded by *D. scoparium* and *C. quitensis* viruses may possess a similar activity. As the cell-to-cell movement of *Potato virus X* depends on silencing suppressor activity of TGB1 protein [33], we speculate that the presumed silencing suppression by vDRB proteins may be related to movement functions of other TCMB proteins. Another possible way of vDRB involvement into virus movement can be hypothesized based on the presence of the N-terminal highly hydrophobic segments likely directing vDRB membrane targeting. The cellular DRB proteins related to vDRB are shown to be incorporated into virus-specific replication membrane compartments [21,22] representing structures often located at the PD orifice and involved in virus cell-to-cell movement [23]. Similarly, the hydrophobic vDRB proteins likely interact with cell endomembranes. Thus, these proteins may be proposed to work in concert with other TCMB proteins to take part in viral dsRNA delivery to and/or retaining in the PD-associated ER membrane-derived replicative compartments [8,24]. In any case, irrespective of the mechanism involved, the proposed functional links between vDRB and proteins directly mediating virus cell-to-cell transport would explain conservation of vDRB in TCMB of viruses infecting distant host species. Additionally, as the cellular vDRB-related proteins of DSRM_AtDRB family take part in RNA silencing pathways, including those involving antivirus defense [14,15], and dsRNA-triggered immunity against viruses [16,21], vDRB may conceivably act a dominant-negative counterpart of respective cell proteins, therefore inhibiting their functions in plant antivirus responses.

An important clue to the pathways of evolutionary origin of TGB, BMB and TCMB, to our mind, can be provided by phylogenetic analysis of their encoded helicases. Evidently, phylogenetic analysis suggests that the BMB and TCMB helicases form a common branch, which is closer to benyvirus TGB helicases and less similar to potex- and hordei-like TGB helicases (Figure 1). Moreover, BMB and TCMB helicases show more sequence identity to beny-like replication helicases than to potex- and hordei-like TGB helicases [4]. Therefore, it can be suggested that a starting event in the evolutionary emergence of BMB and TCMB could be duplication and autonomization of the replicative helicase domain possibly occurring due to template switching during the virus genome replication along with probable non-replicative joining of RNA fragments [34]. Such RNA–RNA recombination likely resulted in the formation of the earliest monopartite and/or multipartite beny-like replicons with an autonomized copy of SF1 helicase. Recombination-dependent scenarios for evolutionary radiation have been also proposed for viruses of the family *Hepeviridae* [35], which together with benyviruses and tetraviruses [36], comprise the order *Hepelivirales* [3]. Taking into account the fact that the currently revealed TCMB-containing viruses infect moss, a primitive land plant, or the geographically long-term isolated flowering plant *C.*
*quitensis* [6], TCMB might be considered as an evolutionary old movement gene module originating in benyvirus-like replicons.

Furthermore, the phylogenetic analysis of TCMB-encoded helicases may shed new light on the evolutionary shift between TGB3-containing and TGB3-lacking transport modules. As suggested earlier, the TGB3 gene could be either acquired by a BMB-like transport module to give rise to a TGB-like module consisting of three genes, or, alternatively, lost by a TGB-like module that resulted in the appearance of a BMB-like transport system employing two MPs [4,5,6]. As (i) TGB3 is present in TCMB, but not in BMB, while both these transport modules are presumed to be evolutionarily old based on the phylogenetic analysis of the helicase domain, and (ii) TGB3 is absent in some viruses with evolutionarily younger transport modules, such as potexviruses, typically having this gene in their TGB, the evolutionary scenario involving the loss of TGB3 can be considered as quite probable. Such loss of the TGB3 gene might occur independently in distant virus groups at different stages of evolution of TGB/BMB-like transport modules. This hypothesis is in agreement with the earlier suggestion that the TGB3 protein may be required for virus transport only in certain plant hosts [37], while in others, where it was non-essential, the TGB3 gene could be eliminated in the course of transport module evolution. It should be noted that the events of acquisition and loss of the TGB3 gene may occur repeatedly in the evolution of TGB/BMB/TCMB. Two ways of TGB3 acquisition can be envisaged, either a duplication of the TGB2 gene and further functional specialization of the TGB2 copy, as suggested earlier [6], or a recombination with cellular mRNA resulting in the acquisition of a non-viral hydrophobic protein gene that can further evolve to perform a function in virus movement. In fact, small ORFs, which are known to emerge de novo in cellular long noncoding RNAs, may serve as a continuous reservoir of variable novel polypeptides serving as a raw material for natural selection [38,39]. Such ORFs for small membrane-bound polypeptides are considered as a preferential subject of adaptive evolution because of the escape of their encoded proteins from degradation or other deleterious interactions in the membrane environment [39], and their genes could be an evolutionary source for TGB3 acquisition. Additionally, a horizontal transfer of the TGB3 gene between different viral genomes cannot be excluded.

The predictions of moss ssRNA-containing beny-like virus infecting *D. scoparium* (this study) and another moss-infecting virus, *Leucodon julaceus associated beny-like virus* [40], raise the question of whether the virus cell-to-cell movement in mosses can occur in a way similar to that in higher plants. The available published data suggest that macromolecular trafficking in mosses is quite similar to that in flowering plants and includes trafficking through plasmodesmata having obvious structural similarity to those in seed plants [41,42], which is in agreement with our finding that the moss viruses code for transport module was typical for viruses of higher plants.

In conclusion, the identification of the TCMB transport system that contains, compared to TGB/BMB, an additional vDRB-encoding gene opens new directions in the experimental analysis of both transport modules of this type and the functions of virus-encoded dsRNA-binding proteins, as well as further phylogenetic analyses aimed at the reconstruction of the evolutionary history of movement proteins encoded by plant virus genomes.

## Figures and Tables

**Figure 1 biomolecules-12-00861-f001:**
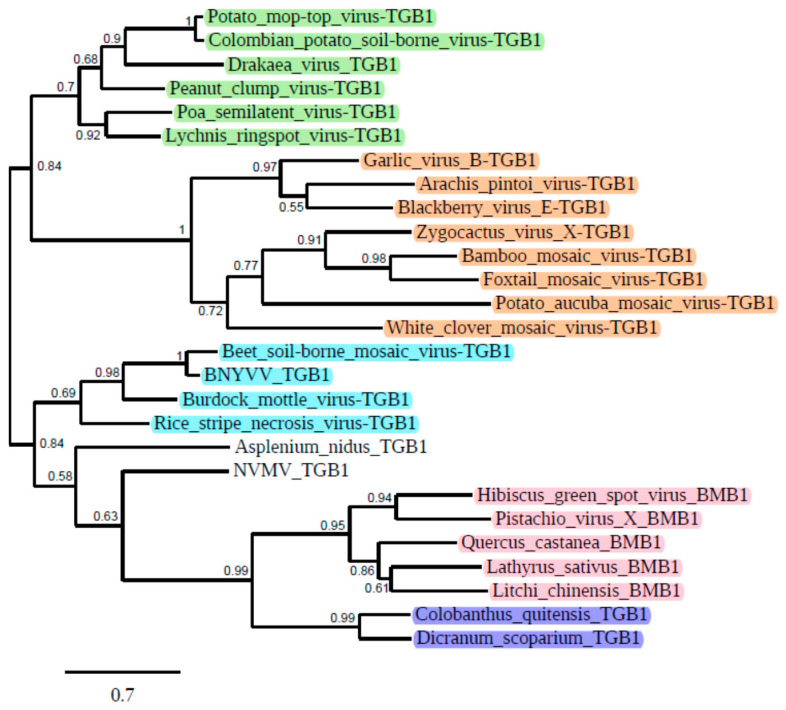
Phylogenetic analysis of the helicase domains derived from the aligned deduced amino acid sequences of the selected proteins encoded by TGBs and BMBs. The phylogenetic unrooted tree was constructed using the maximum likelihood method based on the amino acid sequence alignments (http://www.phylogeny.fr/simple_phylogeny.cgi, accessed on 26 May 2022). The bootstrap values obtained with 1000 replicates are indicated on the branches, and branch lengths correspond to the branch line’s genetic distances. The genetic distance is shown by the scale bar at the lower left. BNYVV—*Beet necrotic yellow vein virus*; NVMV—*Nicotiana velutina mosaic virus*. Hordei-like TGB1 helicases are in green, potex-like TGB1 helicases in brown (including those of *Bamboo mosaic potexvirus* and *Foxtail mosaic potexvirus,* the viruses most close to the potexvirus-type species *Potato virus X*), benyvirus TGB1 helicases are in blue, BMB1 helicases found two viruses and three VLRAs (*Q. castanea*, *L. sativus* and *L. chinesis*) are in pink, TCMB helicases (this paper) found in the corresponding VLRAs (*C. quitensis* and *D. scoparium*) are in dark blue.

**Figure 2 biomolecules-12-00861-f002:**
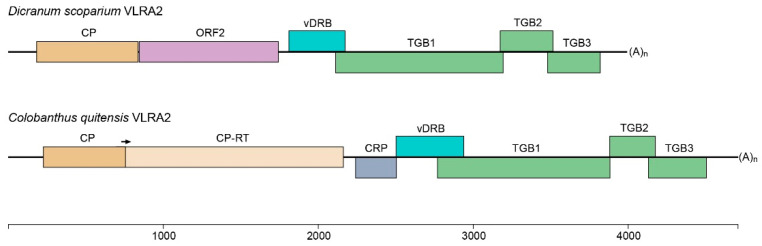
Comparison of gene organization of predicted RNA2 genomic components encoding putative multicomponent cell-to-cell transport systems in *D. scoparium* and *C. quitensis* VLRAs. Genes are shown as boxes with the names of the encoded proteins. Genes of proteins potentially involved in cell-to-cell movement are shown in green (TGB counterparts), light green (vDRB) and blue-green (CRP). Genes encoding small hydrophobic proteins are shown in blue. Replicative genes are shown in yellow. Arrows indicate read-through codons in CP-RT proteins. CRP—cysteine-rich protein, vDRB—virus double-stranded RNA-binding protein, CP-RT—coat protein read-through protein.

**Figure 3 biomolecules-12-00861-f003:**
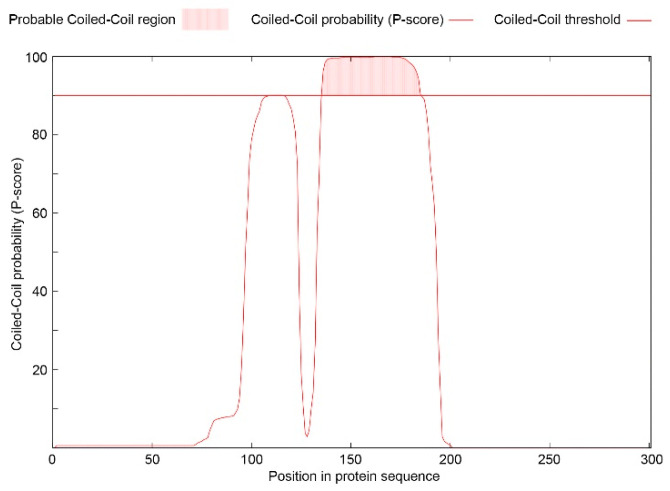
Prediction of coiled-coil segment (above red cut-off line) (https://waggawagga.motorprotein.de/ accessed on 26 May 2022) in the ORF2 protein of Ds-VLRA2.

**Figure 4 biomolecules-12-00861-f004:**
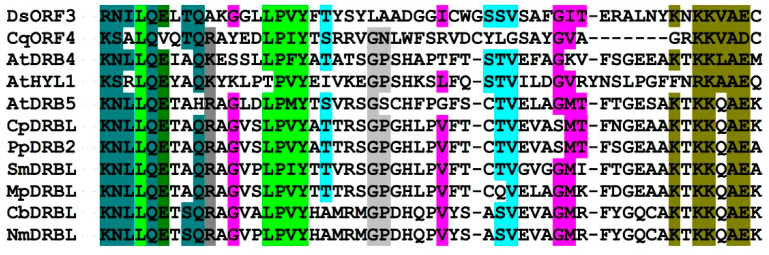
Multiple sequence alignment of the dsRNA-binding active centers of proteins HYL1, DRB4 and DRB5 from *A. thaliana* with vDRB proteins of *D. scoparium* (DsORF3) and *C. quitensis* (CqORF4) as well as DRB-like proteins of moss *Ceratodon purpureus* (CpDRBL—accession KAG0625911), moss *Physcomitrium patens* (PpDRB2-XP_024393530), lycophyte *Selaginella moellendorffii* (SmDRBL-EFJ14280), liverwort *Marchantia polymorpha* (MpDRBL-PTQ26790), charophyte algae *Chara braunii* (CbDRBL-GGXX01036972), charophyte algae *Nitella mirabilis* (NmDRBL-JV799478), charophyte algae *Spirogyra pratensis* (SpDRBL-GFWN01008525). Motifs characteristic of domains of DSRM_AtDRB-like family are shown in color.

**Figure 5 biomolecules-12-00861-f005:**
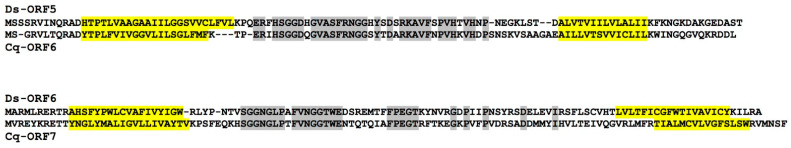
Sequence alignment of small membrane-bound TGB-like proteins encoded by ORF5 and ORF6 in the *D. scoparium* VLRA with those encoded in ORF6 and ORF7 in *C. quitensis* VLRA. Hydrophobic segments are in yellow; residues identical in central hydrophilic regions are shaded. Note consensus “SGG(D/N)xxxxxFxNGGxxxDSRxxxFxPxxTxxN” which is common between the central hydrophilic regions of proteins encoded by Ds-ORF5 and 6.

**Figure 6 biomolecules-12-00861-f006:**
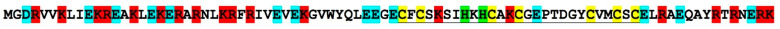
Amino acid sequence of the ORF3 protein encoded by Cq-VLRA2. Positively charged residues are shown in red, negatively charged in blue, cysteine residues in yellow, histidine residues in green. Putative zinc finger motif-containing module is underlined.

**Figure 7 biomolecules-12-00861-f007:**
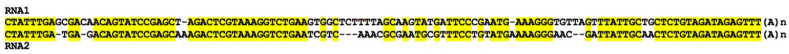
Nucleotide sequence alignment of the 3′-terminal regions preceding poly(A) in the predicted VLRA RNAs 1 and 2 from *Dicranum scoparium*. Highly conserved RNA blocks are highlighted by yellow background.

## Data Availability

Not applicable.

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
