# Peer review of "Bioinformatic Analysis Predicts a Novel Genetic Module Related to Triple Gene and Binary Movement Blocks of Plant Viruses: Tetra-Cistron Movement Block"

_biomolecules, 2022, doi:10.3390/biom12070861_

Round 1

Reviewer 1 Report

This manuscript has been submitted to the Special Issue of Biomolecules "Virology 130 Years After Ivanovsky - A Theme Issue Commemorating the Discovery of Viruses by Dmitri Ivanovsky" and constitutes a very interesting bioinformatics piece of work on plant virus movement proteins (MPs) which control a crucial step in plant virus infection, namely the intercellular transport of viruses. Following the historical context of the Special Issue it is worth noting that the process of plant virus movement has been intensively investigated for more than 40 years (of 130!). However, in spite of such extensive research efforts and some significant but scattered achievements our understanding of the entire process of virus movement still seems to be in its infancy. Interestingly, MPs are unique to plant viruses but show extremely surprising structural and functional diversity while maintaining their core function.  Among activities associated with MPs are modification of plasmodesmata SEL (size exclusion limit), RNA-binding, formation of viral transport complexes or virion tail-like “transport devices”, generation of intercellular tubules, association with viral replication complex, suppression of RNA silencing and pattern triggered immunity (PTI) and some others, which sometimes are poorly integratable even for one and the same virus. Thus, the entire mechanistic picture of virus trafficking is rather far from being complete.

This paper describes bioinformatics prediction suggesting that both virus-like de novo transcriptome assemblies found in moss Dicranum scoparium and Antarctic flowering (but moss-like in appearance) plant Colobanthus quitensis contain novel putative tetra-cistron movement block (TCMB). Based on the sequence similarities to triple (movement) gene block (TGB) and binary (movement) gene block (BMB) of other viruses, the authors suggest that the three downstream putative genes of TCMB encode TGB/BMB-like movement proteins. In addition, the other (upstream) gene of TCMB in both virus assemblies carries sequences that encode a protein with a dsRNA-binding motif (DSRM), which has been designated as the fourth TCMB MP.  

Without going into detail of de novo assembly as well as other bioinformatics methods, I found the “TCMB hypothesis” to be very attractive and quite plausible. I also believe that this paper opens up new directions for future experimental analysis which must support this hypothesis and address the following issues: Do RNAseq assemblies built in this paper in fact represent real viruses that systemically infect their hosts? Are all the TCMB genes really expressed in infected plants? What are real functions of the TCMB proteins, and DSRM, in particular?  Are the specific organization and functions of TCMB adapted and compatible to peculiar cell and tissue morphology/physiology of moss or moss-like flowering plants? By the way, did the authors look at other viruses (transcriptome assemblies) found in moss and their putative movement systems (e.g. Appl Environ Microbiol 84:e01124-18, https://doi.org/10 .1128/AEM.01124-18)? Such information may be particularly important for evolutionary considerations in relating virus movement machineries to genuine plant transport systems, as they may co-evolve with each other. Is anything known about macromolecular trafficking in moss (similarities/differences to typical model plants)?  The authors may wish to address some of these suggestions in the paper.

Minor comments:

1)     As noted above, this is a nice bioinformatics work, but before experimental (biological) validation of the conclusions I would tone down some statements using, for example wording “Novel genetic module has been predicted” instead “Novel genetic module has been identified” and so on. This also relates to the Title – use “putative”.

2)     As indicated in the Introduction, Cq-TGB3 and Cq-TGB3 have been previously shown to exhibit sequence similarity, and the authors believe that that this observation supports the hypothesis of TGB3 origin by duplication of TGB2 gene. However, it remains unclear if the respective putative proteins of Dicranum virus predicted in this paper are also somewhat similar. This point should be clarified, as it is important for discussion of TGB evolution.

3)     The model of TGB evolution presented in the Discussion suggests that the TGB3 gene could be acquired and lost during the evolution, possibly repeatedly. The authors should discuss possible origin of acquired TGB3 genes. Are these genes supposed to be derived from genomes of other viruses or host cell RNAs?

4)     Viral species names (in contrast to virus names) should be written in italics with the first word beginning with a capital letter.

In summary, this interesting paper defines a clear framework for further experimental studies of molecular and cellular functions of TCMB MPs.

Author Response

We thank reviewer for valuable comments. A point-by-point response is below.

1) Is anything known about macromolecular trafficking in moss (similarities/differences to typical model plants)?  

We added the corresponding phrase to the main text. See Discussion p. 10.

2) As noted above, this is a nice bioinformatics work, but before experimental (biological) validation of the conclusions I would tone down some statements using, for example wording “Novel genetic module has been predicted” instead “Novel genetic module has been identified” and so on. This also relates to the Title – use “putative”.

According to referee's comments we made corrections through whole text and, finally, change the paper title.

3) As indicated in the Introduction, Cq-TGB3 and Cq-TGB3 have been previously shown to exhibit sequence similarity, and the authors believe that that this observation supports the hypothesis of TGB3 origin by duplication of TGB2 gene. However, it remains unclear if the respective putative proteins of Dicranum virus predicted in this paper are also somewhat similar. This point should be clarified, as it is important for discussion of TGB evolution.

See main text p. 6 and legend to Fig. 5.

4) The model of TGB evolution presented in the Discussion suggests that the TGB3 gene could be acquired and lost during the evolution, possibly repeatedly. The authors should discuss possible origin of acquired TGB3 genes. Are these genes supposed to be derived from genomes of other viruses or host cell RNAs?

See corrected text in Discussion p 9. and 10.

5) Viral species names (in contrast to virus names) should be written in italics with the first word beginning with a capital letter.

Viral names are modified through whole text.

Reviewer 2 Report

This manuscript begins with a fascinating description of the evolution of TGB and BMB. I think a sentence or two more information is needed for the reader to follow completely. Are these TGB from plant viruses only? I had some confusion after reading the introduction that they originated in plants themselves and were later adapted by viruses. Or only one has been found in the plant bird's nest fern? A little more clarity is needed. 

In Figure 1, where would PVX be in relation to the other viruses? That is one that readers would best recognize. 

The findings are thorough and very interesting. 

Author Response

We thank reviewer for helpful comments.

1) I think a sentence or two more information is needed for the reader to follow completely. Are these TGB from plant viruses only? I had some confusion after reading the introduction that they originated in plants themselves and were later adapted by viruses. Or only one has been found in the plant bird's nest fern? A little more clarity is needed. 

See modified version of Introduction.

2) In Figure 1, where would PVX be in relation to the other viruses? 

See modified legend to fig. 1.